# Molecular basis for heat desensitization of TRPV1 ion channels

Lei Luo[1,2,5], Yunfei Wang[1,2,5], Bowen Li[1,5], Lizhen Xu[3], Peter Muiruri Kamau[1,2], Jie Zheng [4], Fan Yang[3], Shilong Yang[1] & Ren Lai[1]

The transient receptor potential vanilloid 1 (TRPV1) ion channel is a prototypical molecular sensor for noxious heat in mammals. Its role in sustained heat response remains poorly understood, because rapid heat-induced desensitization (Dh) follows tightly heat-induced activation (Ah). To understand the physiological role and structural basis of Dh, we carried out a comparative study of TRPV1 channels in mouse (mV1) and those in platypus (pV1), which naturally lacks Dh. Here we show that a temperature-sensitive interaction between the N- and C-terminal domains of mV1 but not pV1 drives a conformational rearrangement in the pore leading to Dh. We further show that knock-in mice expressing pV1 sensed heat normally but suffered scald damages in a hot environment. Our findings suggest that Dh evolved late during evolution as a protective mechanism and a delicate balance between Ah and Dh is crucial for mammals to sense and respond to noxious heat.

[1] Key Laboratory of Animal Models and Human Disease Mechanisms of Chinese Academy of Sciences/Key Laboratory of Bioactive Peptides of Yunnan Province, Kunming Institute of Zoology, Kunming, Yunnan 650223, China. [2] University of Chinese Academy of Sciences, Beijing 100049, China. [3] Key Laboratory of Medical Neurobiology, Department of Biophysics and Kidney Disease Center, First Affiliated Hospital, Institute of Neuroscience, National Health Commission and Chinese Academy of Medical Sciences, Zhejiang University School of Medicine, Hangzhou, Zhejiang 310058, China. [4] Department of Physiology and Membrane Biology, University of California, Davis, CA 95616, USA. [5]These authors contributed equally: Lei Luo, Yunfei Wang, Bowen Li. Correspondence and requests for materials should be addressed to J.Z. (email: jzheng@ucdavis.edu) or to F.Y. (email: fanyanga@zju.edu.cn) or to S.Y. (email: yangsl@mail.kiz.ac.cn) or to R.L. (email: rlai@mail.kiz.ac.cn)

Many plant and animal species live and thrive in the freezing polar region or the scorching desert. Sophisticated thermoregulatory mechanisms are developed through evolution, especially in highly evolved mammals such as humans. One of the basic elements of these mechanisms is sensing environmental (and internal) temperature. The cloning of TRPV1, a highly temperature-sensitive non-selective cation ion channel[1], allowed investigation of the molecular mechanism of detecting noxious heat (>42 °C). TRPV1 channels are expressed in a subset of peripheral and central neurons where they either directly detect ambient heat or participate in thermoregulation[1,2]. Consistent with these physiological roles, antagonists of TRPV1 reduce heat sensitivity and elevate body core temperature (hyperthermia)[3], whereas a burning sensation and acute drop in body core temperature (hypothermia) are induced by administration of capsaicin, a chemical rich in chili peppers that elicit spiciness by activating TRPV1[1,2].

Following heat-induced activation (Ah), TRPV1 desensitizes (Dh) rapidly (in ms-to-s time scale) if noxious heat persists[4]. Ah has been suggested to arise from conformational changes in several channel structures, including the channel pore[4–8] and both termini[9–11]. However, although the heat-induced Dh has been characterized[12], its underlying biophysical mechanisms and physiological significance remain largely unexplored, partially due to the challenge that Ah and Dh of most mammalian TRPV1 channels are tightly entangled.

To isolate and study Dh, we surveyed TRPV1 channels of diverse animal species for one that may possess naturally separated Ah and Dh. This exploration led us to platypus (*Ornithorhynchus anatinus*) TRPV1 (pV1). Platypus is known to maintain a relatively low body temperature (32 °C) among mammals and is intolerant to ambient temperatures higher than 25 °C[13]. We found most of pV1's functions are normal; the channel is activated by noxious heat, low pH, pungent chemicals, and animal toxins known to activate mouse and human TRPV1. However, pV1 is completely devoid of Dh, providing a unique opportunity to investigate its molecular mechanism and physiological significance. Our investigation at the molecular, cellular, and animal levels revealed that Dh is mediated by an interaction between N and C termini, which is coupled to a conformational rearrangement of the pore domain, leading to selected desensitization of the channel to heat. This process is absent in pV1, leaving pV1 persistently open upon sustained elevation of temperature. Furthermore, we found that Dh acts as a protective mechanism against cell swelling, inflammation, and tissue damage induced by noxious heat.

## Results

**Platypus *trpv1* gene is under positive selection.** Protein molecules, including TRPV1, are expected to gain complexity in their functions during evolution[14,15]. To discover TRPV1 channels with a simpler heat response, we focused on primitive mammals by carrying out evolutionary analyses. Phylogeny of ten species including mammals from lower to higher levels in evolution and a non-mammal species was constructed and shown in the established species tree (Supplementary Fig. 1a). As illustrated in Supplementary Fig. 1b, *trpv1* gene of platypus' genome is highlighted by using the branch-site model (*p*-value: 0.0123)[16], suggesting that this gene may undergo molecular evolution to participate in environmental adaptation. Interestingly, platypus shows special features of heat response, such as a low body core temperature (32 °C), intolerance to high ambient temperatures (>25 °C), and has a fascinating combination of reptilian and mammalian characters of thermoregulation[13], which is consistent with the detected strong signal of selection in its *trpv1* gene.

Therefore, we suspected that heat response of platypus TRPV1 (pV1) may be different from the other mammalian TRPV1 channels.

**pV1 is a polymodal receptor lacking Dh transition.** Electrophysiological analysis revealed that pV1 can be effectively activated by heat (Fig. 1b), such as other mammalian TRPV1 channels. The Ah current of pV1 is robust compared with the current activated by capsaicin (Fig. 1a and Supplementary Table 1). We used supersaturated capsaicin (50 μM) to activate heat-desensitized mV1 (Supplementary Fig. 1c), because these desensitized channels also became less sensitive to capsaicin[12]. The heat activation threshold of pV1, at ~35 °C, is slightly higher than the body core temperature of platypus (32 °C)[17,18] (Fig. 1b, c), again reminiscent of other mammalian TRPV1 channels. In addition, activation of pV1 is polymodal, as the channel can be directly activated by low pH, 2-aminoethoxydiphenyl borate, divalent cations, and RhTx (Supplementary Fig. 1d). However, pV1 currents did not desensitize during prolonged heating (Fig. 1b), a common process of most mammalian TRPV1 channels (Supplementary Fig. 1c). Separation of Ah and Dh suggests that they are driven by distinct gating processes. Therefore, pV1 offers a unique opportunity to investigate the structural mechanism underlying Dh of TRPV1 channels.

**Both N and C termini of mV1 bestows pV1 with Dh.** We took advantage of chimeric constructs to probe channel structures critical for Dh. Replacing the pV1 N terminus or C terminus with the corresponding parts of mV1 failed to introduce discernable desensitization to continuous heating (Fig. 1d and Supplementary Tables 2, 10). We next constructed another chimeric channel (pV1_mNC), in which both N and C termini were transplanted from mV1 to pV1. Surprisingly, we found that pV1_mNC exhibited robust desensitization upon 40 °C heating like the wild-type (WT) mV1 (Fig. 1d, e), with the current declining time constant, τ, being 16.9 ± 0.4 s (mean ± SEM, $n = 5$), similar to that of mV1 (17.3 ± 0.5 s, $n = 5$; $P = 0.55$) (Supplementary Table 2). Our results demonstrate that both N terminus and C terminus participate in Dh. The observation of both Ah and Dh in pV1_mNC suggests that the pV1 transmembrane domain is capable of supporting heat activation and desensitization, and the lack of Dh in pV1 is likely due to its N or/and C terminus.

**Exogenous free C-terminal peptides inhibit Dh of mV1.** We next investigated the molecular mechanism of Dh. As Dh could be restored in pV1 by transplanting mV1 N and C termini but not only the N or C terminus, we hypothesized that a direct interaction between these terminal domains is required for Dh. If this is the case, Dh should be sensitive to exogenous application of free C-terminal peptides through a competition mechanism. Indeed, we found that Dh of mV1 could be largely inhibited by co-expression with the C terminus of mV1, but not that of pV1 (Fig. 1f). We further characterized Dh of mV1 by applying free C-terminal peptides of different lengths through the patch pipette in whole-cell recordings (Supplementary Fig. 1e). We observed that, at 1 mM concentration, a 30-amino-acid long C-terminal peptide could specifically inhibit Dh of mV1 (Fig. 1g). In contrast, a negative control peptide of the same length but a random amino acid sequence failed to inhibit Dh (Fig. 1g). Moreover, when we de novo modeled the distal N and C termini, which were partially missing from the published single-particle electron cryomicroscopy (cryo-EM) structures[6,19,20], we observed that these termini could interact with each other (Supplementary Fig. 2). These results strongly suggest that a direct interaction between N and C termini is required for Dh.

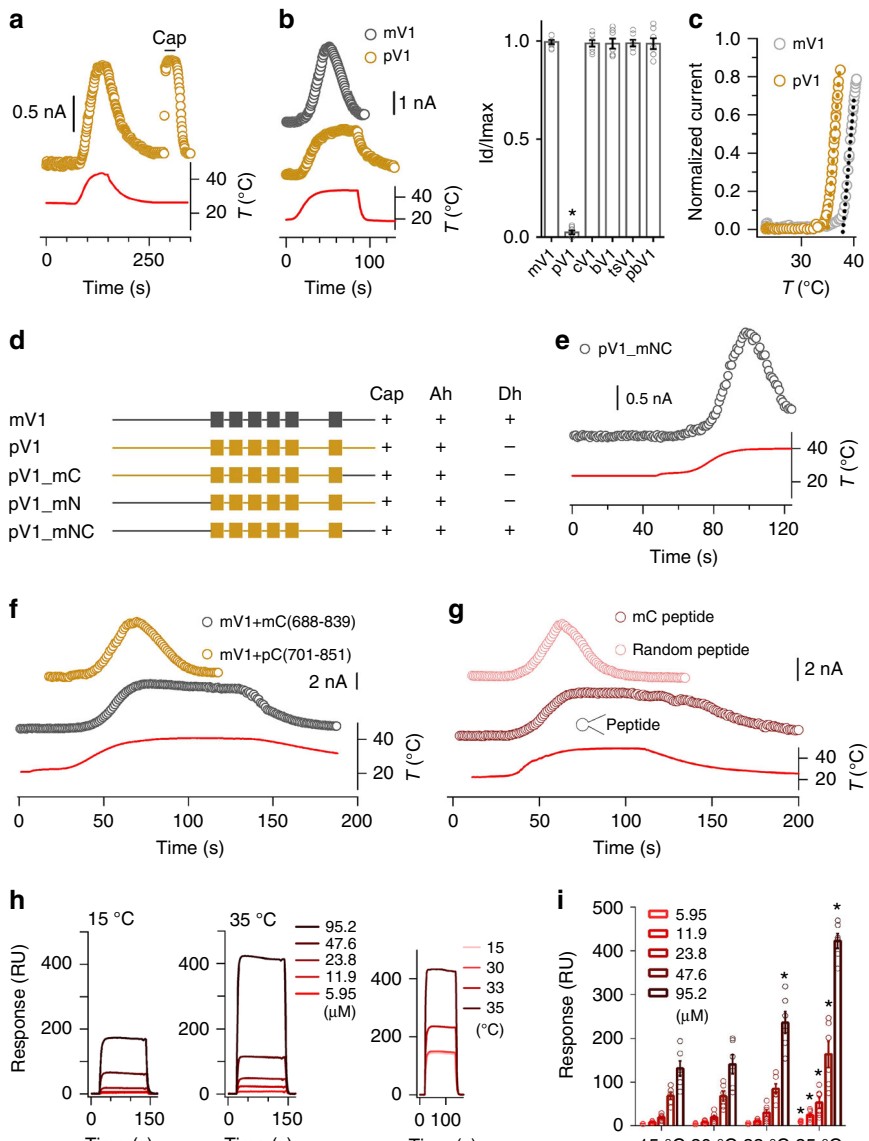

**Fig. 1** The Dh transition of TRPV1 is related to N and C termini. **a** Representative currents of pV1-overexpressing cell activated by heat and 10 μM capsaicin. **b** Example current responses of mV1 (gray) and pV1 (yellow) in response to a temperature ramp (left panel). Note that the desensitization of mV1 occurs before cooling starts. Amplitude ratio (*n* = 7, two-sided *t*-test: **p* < 0.01) between desensitized currents (Id) and maximum currents (Imax) induced by heat at 45 °C (right panel). **c** Superimposed representative temperature–response relationships of heat-evoked currents in HEK293T cells transiently transfected with pV1 and mV1. **d** Chimeric channels were generated between mV1 (gray) and pV1 (yellow) on the basis of the location of transmembrane domains. Their functional evaluation was interpreted as 10 μM capsaicin (Cap), heat-induced activation (Heat/a), and heat-induced desensitization (Heat/d). **e** Current induced by a temperature ramp recorded from HEK293T cells expressing pV1_mNC. **f** Time-dependent Dh of mV1. Channel plasmid was co-expressed with C-terminal region of pV1 (pC, 701N-851E) or mV1 (mC, 688N-839K). **g** Example current induced by a temperature ramp recorded from whole-cell patches with 1 mM mV1 C-terminal peptide (mC peptide) or random peptide in the pipette solution. **h** The mC peptide was perfused over an SPR sensor coated with the N-terminal region of mV1 (mN region, 1M-433R) at 15 or 35 °C (left and middle panels). The mC peptide in 95.2 μM was perfused over an SPR sensor coated with mN region at 15, 30, 33, and 35 °C (right panel). **i** Significant concentration and temperaturedependence of the interaction between N and C termini (*n* = 6, two-sided *t*-test: **p* < 0.01). All statistical data are given as mean ± SEM. Source data are provided as a Source data file

**The interaction between N and C termini is heat dependent**. If Dh is an independent transition separated from Ah as the observations in pV1 indicated (Fig. 1b), the interaction between N and C termini is expected to be thermosensitive. We examined potential N–C interaction using surface plasmon resonance (SPR). N-terminal peptides were immobilized onto a chip, whereas different concentrations of C-terminal peptides (a 30-amino-acid molecule) were first added at 15 °C. At this temperature in the presence of 47.6 μM C-terminal peptides, the SPR signal was ~65 RU. Increasing the concentration of the C-

terminal peptides to 95.2 μM led to a 162% increase in SPR response to ~170 RU. As we expected, at an elevated temperature (35 °C), the affinity between N and C termini was largely increased, as the same increase in C-terminal peptide concentration (from 47.6 to 95.2 μM) caused a 282% increase in the SPR signal (177.9 ± 0.8 to 422.9 ± 0.7 RU) (Fig. 1h, left and middle panel). Indeed, we observed that a steep increase of SPR signal when the temperature is higher than 30 °C (Fig. 1h, right panel and 1i). Moreover, the kinetic equilibrium dissociation constant significantly decreased from 140 nM (at low

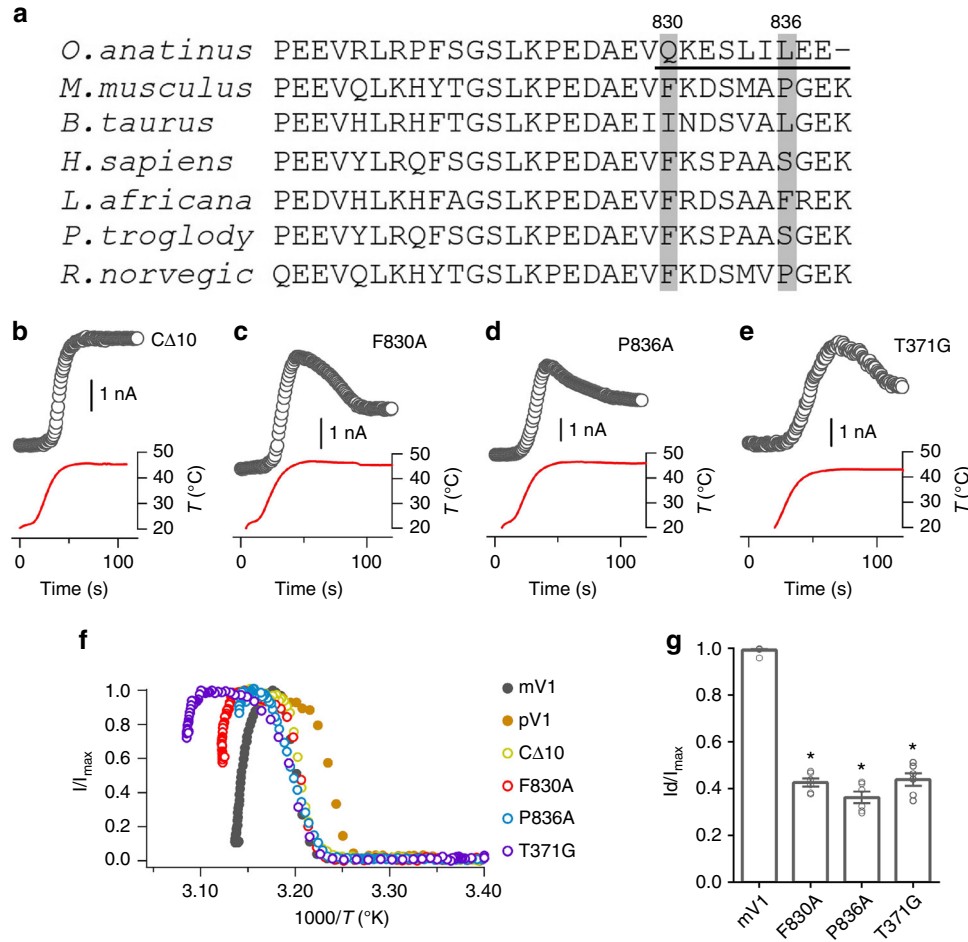

**Fig. 2** N and C termini point mutants exhibit altered Dh transition of TRPV1. **a** The amino acid sequence representing C termini from pV1 is aligned with the corresponding sequences from other six mammal's TRPV1. The species-specific residues are indicated by gray background. **b–e** Representative currents induced by a temperature ramp recorded from HEK293T cells expressing CΔ10 (**b**), F830A (**c**), P836A (**d**), and T371G (**e**) at +80 mV. **f** Example heat-induced current responses from mV1, pV1, and the channel mutants. **g** The desensitized current (Id) of mV1 and mutants is normalized to maximum currents (Imax) induced by heating. $n = 10$ for mV1, $n = 6$ for F830A, $n = 6$ for P836A, $n = 6$ for T371G, two-sided $t$-test: $*p < 0.01$. Data are mean ± SEM. Source data are provided as a Source data file

temperature) to 3 nM (at high temperature) (Fig. 1i), further demonstrating that the interaction between N and C termini is temperature dependent.

**Mutations in N and C termini affects Dh of mV1.** If N and C termini directly interact during Dh of mV1, mutations in these regions that disrupt such an interaction are expected to affect Dh. Sequence alignments of pV1 and other mammalian TRPV1 channels revealed that several amino acids in the C-terminal domain are highly platypus-specific (Fig. 2a). The region harboring these amino acids was previously found to be important for gating[21,22]. We performed deletion and site-directed point mutagenesis targeting these positions (Fig. 2a). Stable temperature control in patch-clamp experiments was achieved to maintain 45 ± 1 °C for 1 min with an optimized perfusion system (Supplementary Fig. 1f). Among all functional truncation mutants (Supplementary Table 3), we found that a ten-amino-acid deletion in the C-terminal domain abolished Dh (Fig. 2b and Supplementary Table 11). To further pin-point the critical residues, we conducted an alanine scan in this region (Supplementary Tables 4 and 12). Interestingly, we observed that an alanine substitution at position 830 or 836 (mV1 number) strongly reduced the degree of Dh from 99.3% of the WT mV1 to 42.7% and 36.3%, respectively (Fig. 2c, d), without affecting the

threshold temperature of Ah (Supplementary Table 4). Fitting the Dh time course to a single-exponential function yielded $\tau$-values of 40.0 ± 0.6 s for F830A mutant channel and 39.9 ± 0.7 s for P836A, compared with 17.3 ± 0.5 s for the WT mV1 (Supplementary Table 3). A previous observation with TRPV1/TRPV3 chimera[23] is also consistent with our interpretation that C-terminal domain contributes to Dh of rat TRPV1. Furthermore, we found T371 in the N-terminal domain contributed to the onset of Dh in a similar way (Fig. 2e and Supplementary Table 4). Our results again support the idea that N and C termini play a crucial role in Dh (Fig. 2f, g).

**The interaction between N and C termini is linked to Dh.** To further test whether the interaction between N and C termini is directly linked to Dh, we simultaneously recorded the time courses of current and fluorescence resonance energy transfer (FRET) signal to directly correlate Dh to conformational changes during N and C interaction. Briefly, we incorporated an unnatural fluorescent amino acid, 3-(6-acetylnaphthalen-2-ylamino)-2-aminopropanoic acid (ANAP), at position 241 in the N terminus of mV1 or pV1; ANAP at the equivalent position of rat TRPV1 did not exhibit any fluorescence change during capsaicin activation[24], making the site a good choice for FRET experiments. We also genetically fused an enhanced fluorescence protein

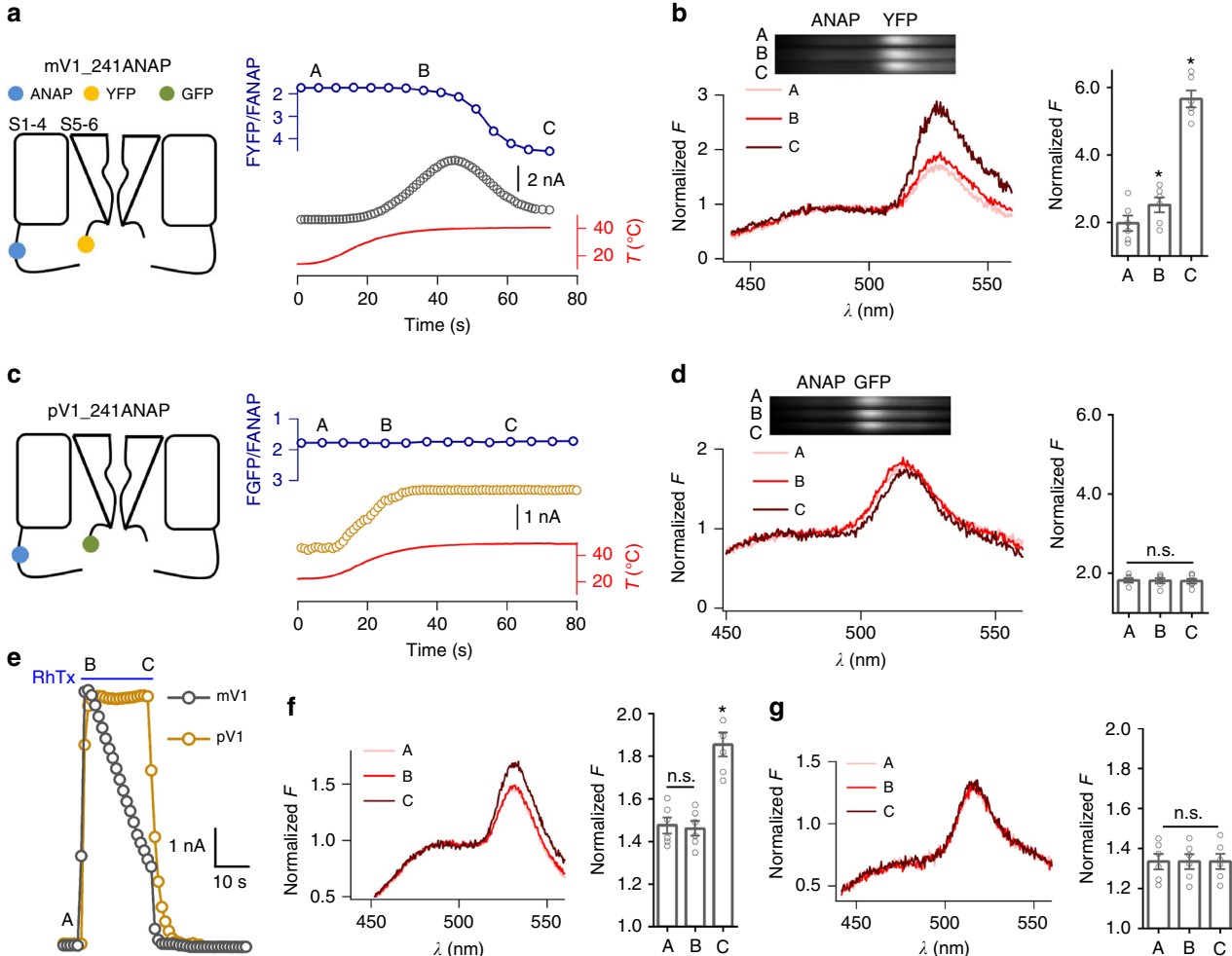

**Fig. 3** Dh transition is required to the N–C direct interaction. **a** A schematic diagram showing the fluorescence-labeled mV1 channel (left panel). Representative current and the YFP/ANAP ratio induced by a temperature ramp were recorded from HEK293T cells expressing mV1_241ANAP. Representative time course of Dh and the corresponding YFP/ANAP ratio recorded from a whole-cell patch are fitted to a single-exponential function (right panel). $\tau$-values of Dh and YFP/ANAP ratio are 12.4 ± 1.4 and 14.9 ± 1.4 s, respectively ($n = 3$). **b** Spectral images of the cell (in **a**) were acquired at points A, B, and C during electrophysiological recording (left panel). The corresponding spectra are normalized to the ANAP intensity and corrected by temperature-dependent FRET signals shown in Supplementary Fig. 3b (right panel, $n = 6$, two-sided $t$-test: *$p < 0.01$). **c** A schematic model of fluorescence-labled mV1 channel (left panel). Simultaneous whole-cell current and fluorescence recordings from pV1_241ANAP when the temperature is changed (right panel). **d** Spectral images of the cell (in **c**) were acquired at points A, B, and C (left panel). The corresponding spectra are normalized to the ANAP intensity and corrected by temperature-dependent FRET signals illustrated in Supplementary Fig. 3d (right panel, $n = 6$, two-sided $t$-test: n.s., not significant). **e** Whole-cell recording from mV1_241ANAP and pV1_241ANAP in the presence of 10 μM RhTx. **f** Fluorescence emission spectra of the cell (mV1 in **e**) were recorded at time points A, B, and C during current recordings (left panel). The corresponding spectra are normalized to the ANAP intensity (right panel, $n = 3$, two-sided $t$-test: *$p < 0.01$; n.s., not significant). **g** Fluorescence emission spectra of the cell (pV1 in **e**) were recorded at time points A, B, and C (left panel). The corresponding spectra are normalized to the ANAP intensity (right panel, $n = 6$, two-sided $t$-test: n.s., not significant). All statistical data are given as mean ± SEM. Source data are provided as a Source data file

(enhanced yellow fluorescent protein (eYFP) and enhanced green fluorescent protein (eGFP) for mV1 and pV1, respectively) to their C-terminal end. All fluorophore-labeled channels exhibited functional properties similar to the WT channels (Supplementary Table 5). FRET was observed between ANAP and fluorescence protein[25]. By whole-cell patch fluorometry recordings[5], we observed in mV1 a substantial increase in FRET when channels transitioned from the Ah state into the Dh state (Fig. 3a, b and Supplementary Table 13). In contrast, no discernible change in FRET was detected in pV1 (Fig. 3c, d and Supplementary Table 13). As fluorescence emission is sensitive to temperature, we first measured the effect of heating on fluorophores in a control FRET system where ANAP was directly incorporated to the N terminus of GFP/YFP, in which the distance between

fluorophores is likely stable during heating (Supplementary Fig. 3a–d). We have employed these measurements to correct our FRET imaging of ANAP-incorporated mV1 and pV1 (Fig. 3b, d).

To further corroborate our findings in FRET experiments of heat-desensitized channels, we took an alternative experimental strategy where we measured the changes in FRET induced by centipede toxin RhTx. Previous studies[26,27] have shown that peptide toxins such as RhTx and DkTx mimic heat activation of TRPV1. Indeed, we observed that RhTx not only activated mV1 but also desensitized the channel upon perfusion, but in pV1 where the heat desensitization was absent RhTx also failed to desensitize the channel (Fig. 3e). Thus, we believe that the prolonged application of RhTx mimicked the effect of sustained heating. Using RhTx as a tool, we observed that just like heat,

RhTx induced significant changes in YFP/ANAP ratio (therefore, FRET efficiency) in mV1, but not in pV1 lacking the Dh process (Fig. 3f, g). Therefore, based on our observations from the ANAP-YFP/GFP constructs and RhTx-induced FRET changes, we believe that the FRET increase observed upon sustained heating truly reflected conformational rearrangements during the Dh process in mV1, which lead us to conclude that the interaction between N and C termini is directly linked to Dh. Moreover, our observation that no FRET change occurred during heat activation is consistent with the notion that N and C termini are not required for Ah[28].

**The N–C interaction is coupled to the outer pore domain**. As desensitization yields a closed channel pore, we next asked whether the N–C interaction is coupled to a conformational change in the pore domain during Dh. Given that changes in local hydrophobicity shift the emission spectrum of ANAP[25,29], we employed ANAP again to explore conformational dynamics in the pore domain. There was no change of the emission spectrum of free ANAP during heating (Supplementary Fig. 3e, f). We first labeled ANAP at position 579 on the S4–S5 linker, which was previously used to report a conformational change during capsaicin-induced activation[25]. During Ah, similar to the observation in capsaicin-induced activation[25], we detected a red shift of the emission spectrum from 488 to 495 nm (Supplementary Fig. 3g, h), suggesting an expansion of the lower gate to open the channel. Besides such a change in ANAP emission during Ah, we observed an additional red shift during the Dh transition (Supplementary Fig. 3g, h). The continuous expansion of the lower gate during Dh indicates that constriction to ion permeation in the Dh state is unlikely caused by the lower gate. In contrast, the emission spectrum of ANAP incorporated at position 630 remained unchanged during heat activation but shifted from 492 to 481 nm when the channels were trapped in the Dh state, suggesting a Dh-specific conformational change in the outer pore region. Among six functional channel mutants we have tested (Supplementary Tables 5 and 13), only ANAP at position 630 had a discernable shift of the emission spectrum during the Dh transition. Importantly, no shift of the emission spectrum could be detected with ANAP at the equivalent position of the Dh-deficient pV1 (Fig. 4b).

Based on the site-specific structural dynamics information, we computationally modeled the three-dimensional structure of mV1 in the Dh state. The blue shift in ANAP emission indicated a transition into a more hydrophobic environment at the 630 site, which provided an experimentally derived constraint for structural modeling using a recently published approach[25]. As the heat-activated state of TRPV1 channels has not been resolved, we employed the double-knot toxin-bound state determined by cryo-EM (PDB ID: 3J5Q)[20] as the open state model, because a previous study had shown that Redhead toxin bound to an adjacent outer pore site of TRPV1 uses the heat activation machinery to open the channel[26].

In our Dh state model, we observed that the selectivity filter showed discernable conformational changes as compared with the open state (Fig. 4d and Supplementary Fig. 4). The selectivity filter in each subunit moved closer, leading to a much-reduced pore radius (0.70 Å, compared with 3.16 Å in the open state) in this region (Fig. 4e). The Y672 residue in the middle of the S6 helix has been reported to form a restriction site for ion permeation[30]. We also found that the S6 helix exhibited conformational changes, which made the pore radius at the Y672 residue (0.28 Å) too narrow to permeate ions (Fig. 4e). The ion conductance of our Dh state model predicted by the HOLE program was similar to that of cryo-EM structure determined at

the closed state (Fig. 4f) (PDB ID: 3J5P), suggesting that the conformation rearrangements in the pore region as we observed above indeed lead to a desensitized, therefore closed state.

**Physiological deficiency in platypus *trpv1* knock-in mice**. The absence of Dh in pV1 also provided a unique opportunity to examine the physiological significance of Dh in heat response, which remains unclear as existing *trpv1*-edited mice loss Ah and Dh simultaneously[2,31]. We hereby established platypus *trpv1* gene knock-in mice to functionally replace mouse TRPV1 (we named p-*trpv1* mice; Supplementary Fig. 5a, b and Supplementary Table 6), which showed normal physiological characteristics in urine and blood tests (Supplementary Tables 7–9). The transcription levels of TRPV1 and other channels known to be involved in heat sensing were unchanged in the p-*trpv1* mice (Supplementary Fig. 5c, d). We confirmed using patch clamping that the pV1-related functions were well maintained in small diameter dorsal root ganglion (DRG) neurons of the p-*trpv1* mice, including the responses to capsaicin and heat (Supplementary Fig. 5e). Importantly, Dh was not observed in DRG neurons of p-*trpv1* mice (Supplementary Fig. 5e), which is consistent with our observations in transiently transfected cells (Fig. 1b). We examined the response of both WT and p-*trpv1* mice to noxious heat. In tail-flick and hot-plate tests, both WT and p-*trpv1* mice exhibited similar heat latency at ambient temperatures over 40 °C (Fig. 5a, b). This is consistent with the finding that responses to noxious heat are mediated by likely multiple heat sensors instead of just TRPV1[31].

Interestingly, we observed that the p-*trpv1* mice exhibited constant heat avoidance behavior as they kept walking on the hot plate at 45 °C (Fig. 5c). In contrast, WT mice gradually decreased their movement within 30 min (Fig. 5c), indicating sensory adaptation. Moreover, we found that repetitive hot-plate assays elicited obvious scald injury in the paws of p-*trpv1* but not WT mice (Fig. 5d), which was clearly identifiable by histological examination of degradation and inflammation in paw tissue slides (Fig. 5e, f). Importantly, knocking out *trpv1* alleviated the symptom of scald injury (Fig. 5g) even though the mice could still sense noxious heat[2,32], suggesting that continuous activation of TPRV1 contributed to the scald injury. We tested the dynamics of four cytokines in the paw tissue. In agreement with histological examinations, significant increases in tumor necrosis factor-α, interleukin (IL)-1β, IL-6, and IL-8 were observed in p-*trpv1* but not WT or knockout (KO) mice (Fig. 6a). Apoptosis was not detected during the degradation of paw tissue, given that the biomarker of apoptosis was not upregulated (Fig. 6b and Supplementary Fig. 6). However, pV1-overexpressing HEK293T-cells exhibited much higher rates of swelling/death under heat stress compared with mV1-overexpressing cells (Fig. 6c). Therefore, our results indicate that in highly evolved mammals, TRPV1 channels acquired Dh as a protective mechanism against noxious heat by preventing cell swelling/death of TRPV1-expressing cells and lowering the risk of scald injury.

### Discussion
Our observations from complementary experimental approaches at the molecular, cellular, and system levels revealed that Ah and Dh are separable gating processes in TRPV1 channels, and the Dh process serves as a protective mechanism against tissue damage caused by noxious heat. Particularly, our findings present an underlying structural model for the previous observations of Dh phenomenon, in which a coupled Ah-Dh gating process was described[12]. Dh is tightly associated with the temperature-sensitive dynamic interaction between N and C termini domains. Both domains are known to be the target of important

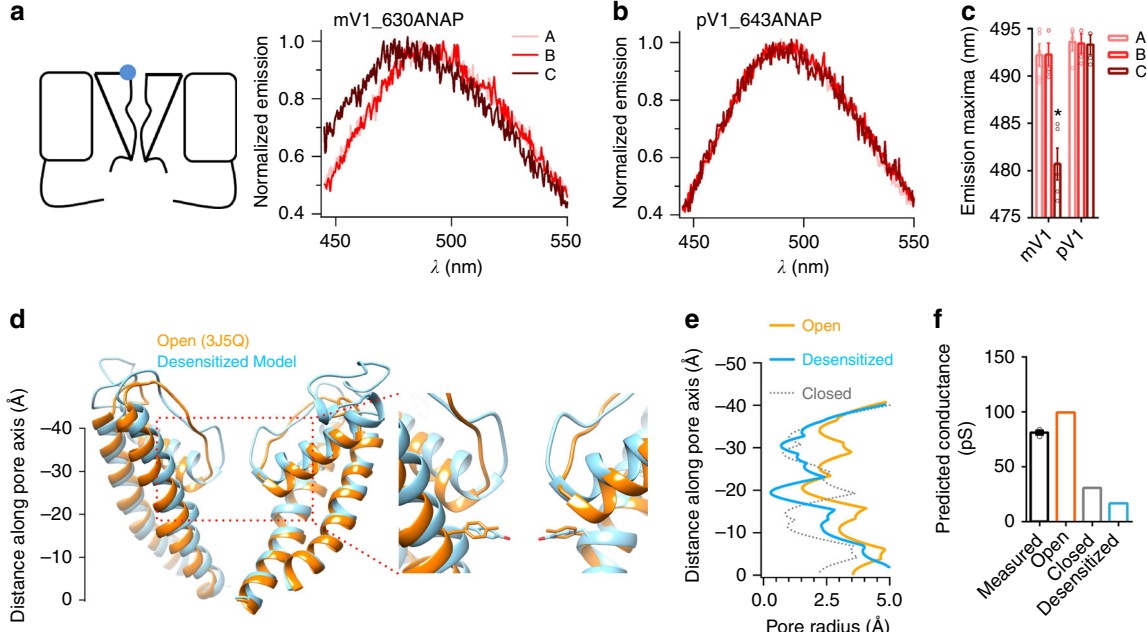

**Fig. 4** Dh transition leads constriction of selective filter. **a** A schematic diagram showing the fluorescence-labeled mV1 and pV1 (left panel). Molecular dynamics simulations of ANAP-labeled mV1 (mV1_630ANAP) at points A, B, and C during electrophysiological recording shown in Fig. 3a. **b** Molecular dynamics simulations of ANAP-labeled pV1 (pV1_643ANAP) at points A, B, and C during electrophysiological recording shown in Fig. 3c. **c** Summary of the emission maxima changes of ANAP labeled in mV1 ($n = 5$) and pV1 ($n = 4$), two-sided $t$-test: *$p < 0.01$. **d** Structural alignment of TRPV1 channel in the open state (orange, PDB ID: 3J5Q) and our model of the desensitized state (blue). The selectivity filter and the Y671 residue exhibited discernable conformational rearrangements between these two states (dashed box in red). **e** Distribution of pore radii in the cryo-EM-derived open state (3J5Q) (line in orange), our modeled desensitized state (line in blue), and the closed state (3J5P) (line in gray). Pore radii were calculated by the HOLE program[50]. **f** Conductance of TRPV1 in different states predicted by the HOLE program. Our desensitized state model (bar in blue) was predicted to have similar conductance as the closed state (3J5P) (bar in gray). The open state (3J5Q) (bar in orange) shows a predicted conductance close to our experimentally measured values[55]. Data are mean ± SEM. Source data are provided as a Source data file

modulators of TRPV1 channels such as ATP and calmodulin[33,34]. Although complex formation by isolated N and C termini domains was not observed biochemically[34,35], cryo-EM structures of TRPV1 proteins suggested that these termini are tightly packed[6,19,20]. In the present study, site-directed incorporation of ANAP allowed us to perform FRET imaging in live cells to assess the interaction between N and C termini. Combination of electrophysiological and pharmacological methods revealed that (i) the increase of FRET efficiency between these domains was only seen during Dh but not Ah (Fig. 3a, b); (ii) the kinetics of FRET increase matched that of current decline (Fig. 3a); (iii) no change of FRET efficiency was detected during heat-induced gating of pV1, in which Dh is naturally absent (Fig. 3c, d); (iv) disruption of N–C interaction by competition of co-expressed or perfused free C-terminal peptides only affected Dh but not Ah (Fig. 1e, f). Together, these observations showed that a state-dependent interaction between N and C termini not only exists in TRPV1 under physiological conditions but also is a prerequisite for Dh.

The N and C termini interaction provided us with an opportunity to study the long-range allosteric coupling mechanism in ion channels. When these intracellular domains interact during Dh, the pore of TRPV1, which locates within the plasma membrane at least 30 Å above[19], is brought to a closed state to desensitize the channel. Such long-range couplings have been reported in many channels such as TRPM2 channel[7], CNG channel[36], HCN channel[37], and BK channel[5], where ADPR, cAMP, and calcium ions, respectively, bind to the intracellular domains to trigger pore opening. Based on spectroscopic imaging of ANAP labeled at specific sites and computational modeling constrained with the experimentally derived information, we suggest that Dh leads to closure of the channel pore at or/and near the selectivity filter, but not at the S6 bundle crossing.

The allosteric nature in the gating of TRPV1 channels largely defines the physiological functions of these channels[26]. Although previous studies with KO mice have demonstrated that TRPV1 contributes to thermo-sensing physiology in mammals[2,31], a tight coupling of the Ah and Dh processes leads to opacity in interpreting observations in animal studies[17,38]. This is clearly illustrated by the transgenic p-trpv1 mice. Functional prolongation of the Ah process in vivo by removing Dh revealed that desensitization serves as a crucial protective mechanism for mammals living in hot environments. In this sense, TRPV1 desensitizing to heat serves a function similar to voltage-gated $Na_V$ and $K_V$ channels desensitizing to depolarization and ligand-gated channels desensitizing to prolonged presence of ligand molecules[26,39]. Desensitization is a general feature of all sensory modalities; it enables acute detection of a much wider range of stimulus intensity and duration. For TRPV1, Dh serves an additional protective role to its host cells, which is required for advanced mammals to explore and survive harsh environments but is not yet acquired by platypus limited to living in temperatures below 25 °C. Our findings on Dh illustrate a critical protective mechanism against noxious heat in mammals.

## Methods

**Ethics statement**. All the animal experiments were carried out in strict accordance with recommendations in the Guide for the Care and Use of Laboratory Animals of Kunming Institute of Zoology, Chinese Academy of Sciences. Protocols were approved by the Institutional Animal Care and Use Committees at Kunming Institute of Zoology, Chinese Academy of Sciences (approval ID: SMKX2016023).

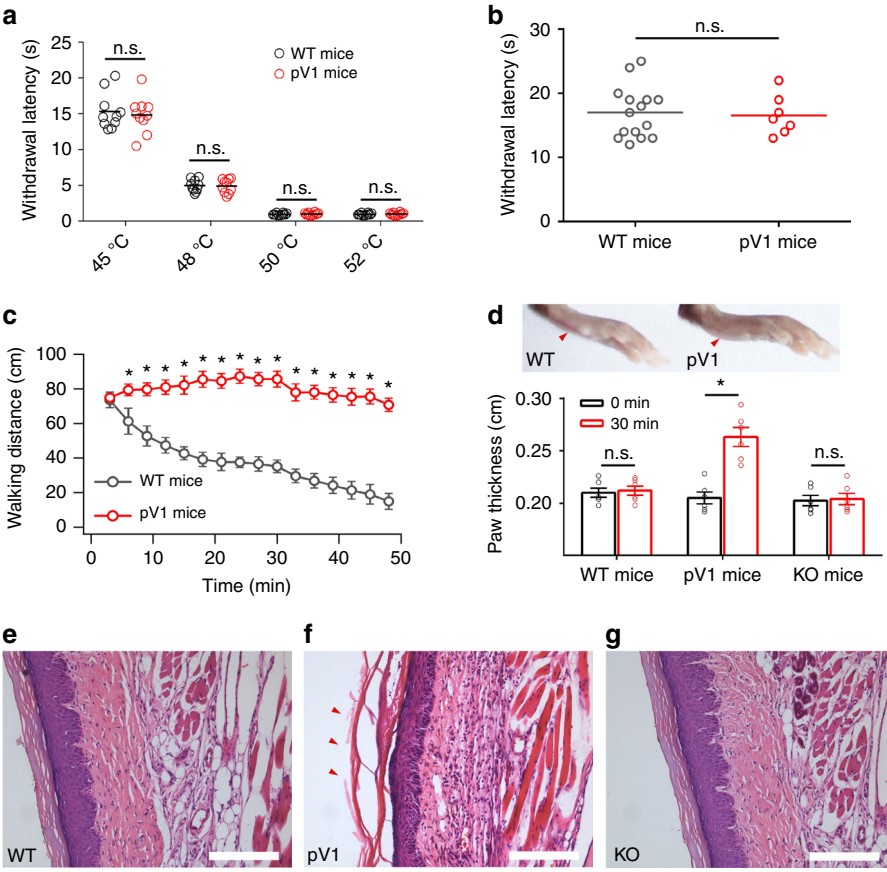

**Fig. 5** Dh transition provides a feedback and protective mechanism against scald damages. **a**, **b** Withdrawal latencies of female mice in the tail-flick ($n = 10$, two-sided $t$-test: n.s., not significant) (**a**) and hot plate ($n = 15$ for WT mice, $n = 7$ for p-trpv1 mice, two-sided $t$-test: n.s., not significant) (**b**) assays. **c** Walking distance for every 3 min (50 min video-tracking) of female WT ($n = 5$) and p-trpv1 ($n = 3$) mice in the hot-plate assay, two-sided $t$-test: *$p < 0.01$. **d** Images of the left hind paw after six times of hot-plate assays. Red arrows indicated the phenotype of paw thickness of female mice (inset). Comparison of left hind paw thickness before and after hot-plate assays ($n = 6$, two-sided $t$-test: *$p < 0.01$; n.s., not significant). **e–g** Photomicrographs of the left hind paw sections taken from female WT (**e**), p-trpv1 (**f**), and trpv1-KO (**g**) mice after hot-plate assays. Note that paw thickness and tissue sections are measured from left hind paw tissue of female mice before and after six times of hot-plate assays. Scale bar, 100 μm. All statistical data are given as mean ± SEM. Source data are provided as a Source data file

All possible efforts were employed to reduce the number of animals used and also to minimize animal suffering.

**Animals**. Platypus *trpv1* knock-in mice (p-trpv1 mice) in the C57BL/6L background were generated by the Cyagen Biosciences, Inc. (Guangzhou, China). Briefly, construction of the platypus *trpv1* allele was based on gene targeting whereby a complementary DNA containing platypus *trpv1* with a loxP-flanked *neo* cassette was targeted to the locus encoding TRPV1 using homologous recombination. Both polymerase chain reaction (PCR) and Southern blotting approaches were used to verify the targeting to the TRPV1 locus. Targeted embryonic stem cell clones were used to produce chimeric animals using eight-cell stage injection. The knock-in male mice with a neomycin selection cassette were mated with Cre females to obtain offsprings and then inbred to develop the p-trpv1 homozygous mice and their WT littermates. The p-trpv1 homozygous mice without the *neo* cassette, referred to as platypus *trpv1* knock-in mice, were used for all experiments in this study. WT C57BL/6L mice, aged 10–14 weeks, were used in control groups. Animals were genotyped using primers as follows: forward primer: 5′-GAACACC GCCTTGCAGTATTTAC-3′ and reverse primer: 5′-CACTGTAGACAAACATGA AGCGAC-3′. Female mice were used for behavioral experiments unless stated otherwise. DRG neurons were obtained from both male and female mice. Mice were housed in a conventional facility at 21 °C on a 12 h light–dark cycle with unrestricted access to food and water.

**Phylogeny analysis**. The genomic data used in this experiment were derived from Ensemble (http://asia.ensembl.org/index.html, version91).

The OrthoMCL method was employed to identify gene family of the selected study species. In brief, the alternative splicing of each gene was first filtered out and the longest transcripts were retained. Then, all protein sequences of the selected species were performed using Blastp alignment (*E*-value ≤ $1e − 5$), while putting other parameters into default. Finally, to obtain the homologous gene family for each species, the Markov model clustering algorithm was applied.

From the gene family results described above, a phylogenetic tree of the study species was constructed using the single-copy genes, whereby 5199 single-copy gene families were selected. To start with, Mafft software was used to compare the protein sequences of the 5199 single-copy family genes and convert the protein comparison into coding sequence (CDS) comparison. Second, the poorly aligned regions were filtered out using Gblocks with an aim of obtaining a better CDS file. Finally, the GTRGAMMA model of RaxML method was used for analysis. We set the Bootstrap to 100 and used *Xenopus tropicalis* as an outgroup species.

**Molecular biology**. Mouse TRPV1 (a gift from M.X. Zhu, University of Texas Health Science Center at Houston, Houston, TX) and human TRPV1 were used in this study. Platypus *trpv1* was synthesized by Tsingke (Beijing, China) based on the predicted sequence from the wild Duckbill platypus (*O. anatinus*) genome (AAPN00000000.1). The platypus *trpv1* (100077591) sequence was obtained by automated computational analysis of the full *O. anatinus* genome. eGFP was fused to the C terminus of TRPV1 to help identify channel-expressing cells. Fruit bat (*Carollia brevicauda*) *trpv1* (JN006859.1), tree shrew (*Tupaia belangeri chinensis*) *trpv1* (102474103), polar bear (*Ursus maritimus*) *trpv1* (103667135), and camel (*Camelus ferus*) *trpv1* (102515699) orthologs were synthesized by Tsingke based on the predicted gene sequence and subcloned into the pEGFP-N1 vector.

Point mutations were generated by Fast Mutagenesis Kit V2, (SBS Genetech, Co., Ltd, China). All channel mutants were confirmed by DNA sequencing. Mouse and platypus TRPV1 chimeras used in this study were generated by the overlapping extension method[7] and confirmed by DNA sequencing. Briefly, to generate pV1_mNC, the primer pair (5′-CTGATGGGGGAGACAGTGAACAAGATTGCACAAGAGAGC-3′ and 5′-GAAGTTGAAGTAAAATATGCGCTTGACAAATCTGTCCCAC-3′) for mouse TRPV1 and the primer pair (5′-

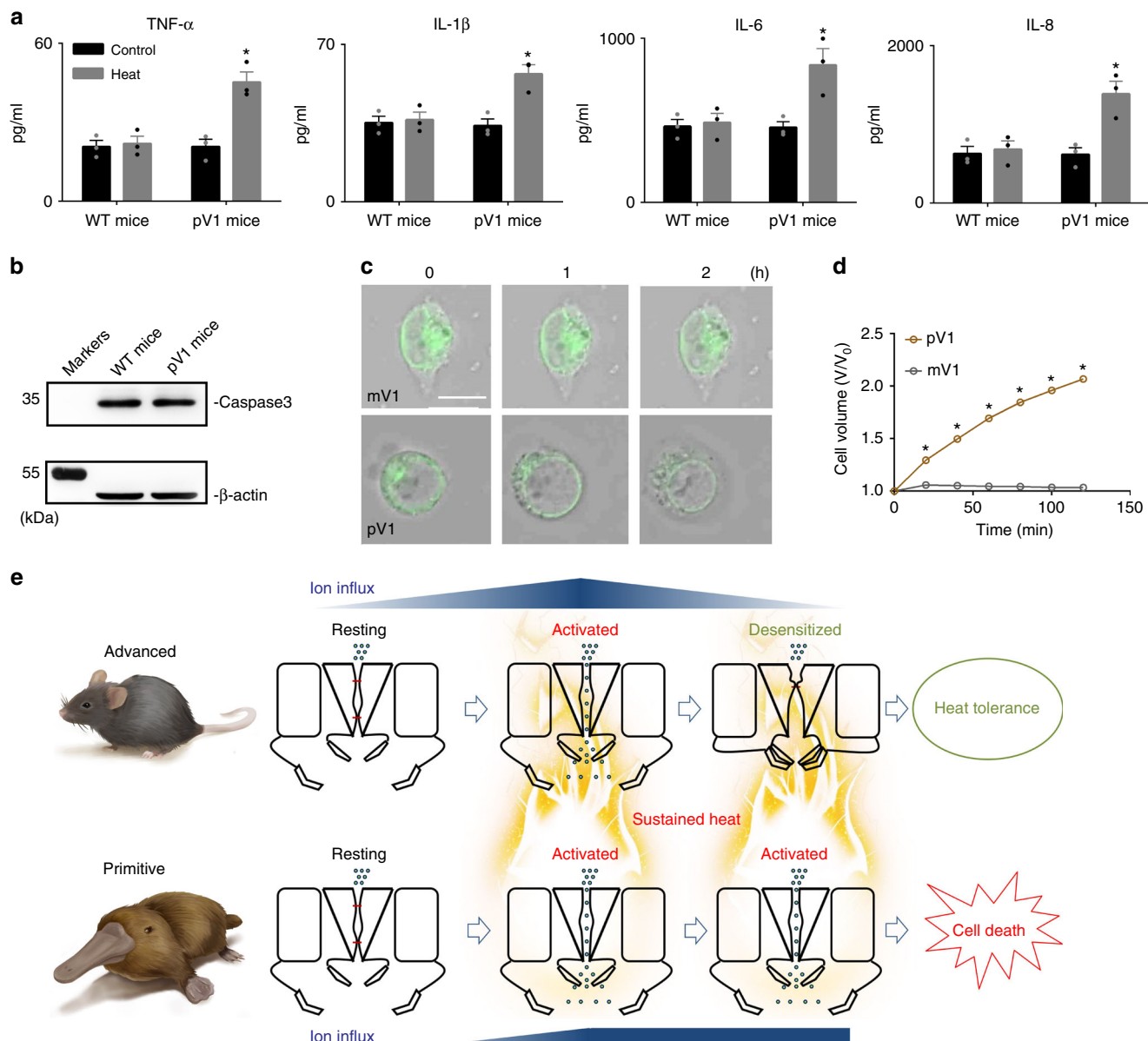

**Fig. 6** Dh transition avoids inflammation and cell death during sustained heat. **a** Levels of TNF-α, IL-β, IL-6, and IL-8 in the left hind paw tissue before and after hot-plate assays ($n = 3$, two-sided $t$-test: *$p < 0.01$). **b** Western blotting showing the absence of apoptotic-like damage in the paw tissue of p-trpv1 mice. **c** Time-lapse microscopy images indicating the swelling of pV1-positive cell body at 40 °C heating. mV1-positive cell body was intact at 40 °C. Scale bar, 10 μm. **d** The time-dependent cell volume of mV1 ($n = 20$) and pV1 ($n = 20$)-positive HEK293T cells, two-sided $t$-test: *$p < 0.01$. Note that cytokines and western blotting are measured from the left hind paw tissue of female mice before and after six times of hot-plate assays. **e** A cartoon illustrating Dh's structural mechanisms and role in mammals against sustained heat. All statistical data are given as mean ± SEM. Source data are provided as a Source data file

GACAGATTTGTCAAGCGCATATTTTACTTCAACTTCTTC-3′ and 5′-CTCTTGTGCAATCTTGTTCACTGTCTCCCCCATCAGGGC-3′) for platypus TRPV1 were used. pV1_mC was generated by the primer pair (5′-CTGATGGGGGAGACAGTGAACAAGATTGCACAAGAGAGC-3′ and 5′- CCGGGCCCGCGGTACCGTTTTCTCCCCTGGGGCCATGGA-3′) for mouse TRPV1 and the primer pair (5′-ATGGCCCCAGGGGAGAAAACGGTACCGCGGGCCCGGGAT-3′ and 5′-CTCTTGTGCAATCTTGTTCACTGTCTCCCCCATCAGGGC-3′) for platypus TRPV1. pV1_mN was generated by the primer pair (5′-GATCTCGAGCTCAAGCTTATGGAGAAATGGGCTAGCTTA-3′ and 5′- GAAGTTGAAGTAAAATATGCGCTTGACAAATCTGTCCCA-3′) for mouse TRPV1 and the primer pair (5′-GACAGATTTGTCAAGCGCATATTTTACTTCAACTTCTTC-3′ and 5′- GCTAGCCCATTTCTCCATAAGCTTGAGCTCGAGATCTGA-3′) for platypus TRPV1.

To generate the mC plasmid, the pEGFP-N1 vector enzyme cutting sites by HindIII and EcoRI were made by QuickCut Hind III and QuickCut EcoR I. The primer pair (5′- GCTAGCGTTTAAACTTAAAACAAGATTGCACAAGAGAGC-

3′ and 5′- CTGGATATCTGCAGAATTTTTCTCCCCTGGGGGCCATGGA-3′) for mouse TRPV1 was generated. pEGFP-N1 enzyme digestion vector and mouse TRPV1 PCR fragment were used for homologous recombination.

The pC plasmid was generated by the primer pair (5′-GCTAGCGTTTAAACTTAAAACAAGGTCTCGCAAGAAAGC-3′ and 5′-CTGGATATCTGCAGAATTTTCTTCTAAAATAAGTGACTC-3′) and the homologous recombination between the pEGFP-N1 enzyme digestion vector and platypus TRPV1 PCR fragment were used.

The C-terminal peptides were synthesized[26] as the following sequences:
Pep_30: PEEVQLKHYTGSLKPEDAEVFKDSMAPGEK
Pep_25: LKHYTGSLKPEDAEVFKDSMAPGEK
Pep_20: GSLKPEDAEVFKDSMAPGEK
Random peptide: QVEEPTYHKLPKLSGVEADEMSDKFKEGPM

**Cell preparation**. HEK293T cells were purchased from Kunming Cell Bank, Kunming Institute of Zoology, Chinese Academy of Sciences (ATCC, CRL-3216). Cells were cultured in Dulbecco's modified Eagle's medium (DMEM) plus 10%

fetal bovine serum with 1% penicillin/streptomycin, incubated at 37 °C in 5% $CO_2$. Cells were transiently transfected with cDNA constructs by Lipofectamine 2000 (Life Technologies, USA) following manufacturer's protocol. One to two days after transfection, electrophysiological recordings were performed.

**Electrophysiology**. All recordings were performed by employing a HEKA EPC10 amplifier with the PatchMaster software (HEKA)[40]. Both pipette solution and bath solution contained 130 mM NaCl, 3 mM HEPES, and 0.2 mM EDTA (pH 7.4). The membrane potential was held at 0 mV and the currents were evoked at ±80 mV (500 ms each). Patch pipettes were prepared from borosilicate glass and fire-polished to resistance of ~4 MΩ. Whole-cell recordings were used to test whether an ANAP-incorporated channel was functional. For whole-cell recording, serial resistance was compensated by 60%. Current was sampled at 10 kHz and filtered at 2.9 kHz. All recordings were performed at room temperature.

Capsaicin was perfused to membrane patch by a gravity-driven system (RSC-200, Bio-Logic). Bath and capsaicin were delivered through separate tubes to minimize the mixing of solutions. Patch pipette was placed in front of the perfusion tube outlet.

**Temperature control**. Temperature control was achieved by perfusion of pre-heated or precooled solutions. Solutions were heated with an SHM-828 eight-line heater controlled by a CL-100 temperature controller (Harvard Apparatus). A custom-made manifold was attached to the output ports of the heater to deliver solutions to the recording chamber and provide heat insulation. Solutions were cooled by embedding the solution reservoirs in ice water and then perfused through a separate line. The patch pipette was placed about 1 mm from the solution output ports. A TA-29 miniature bead thermistor (Harvard Apparatus) was placed right next to the pipette, to ensure accurate monitoring of local temperature. The thermistor's temperature readout was fed into an analog input of the patch amplifier and recorded simultaneously with current. With this method, we achieved rapid and reliable temperature changes.

**Fluorescence unnatural amino acid**. L-ANAP methyl ester was purchased from AsisChem. pANAP vector was purchased from Addgene. ANAP was incorporated into TRPV1 with a TAG amber stop codon mutation[25]. Briefly, after co-transfection of both TRPV1 and pANAP vectors, ANAP was directly added to the culture medium to the final concentration of 20 μM. After 1–2 days, ANAP-containing culture medium was completely changed. Cells were further cultured in ANAP-free medium overnight before experiments.

ANAP fluorescence was excited by the Ar laser with a 375/28 excitation filter, T400lp dichroic mirror, and 435LP emission filter on an inverted fluorescence microscope (Nikon TE2000-U) using a ×40 oil-immersion objective (numerical aperture (NA) 1.3). Emission spectrum of ANAP was imaged with an Acton SpectraPro 2150i spectrograph in conjunction with an Evolve 512 EMCCD camera. The emission peak value was found by fitting the spectrum with a skewed Gauss distribution.

**Site-direct fluorescence recordings and FRET quantification**. For fluorescence recordings, eGFP/eYFP was linked to the C terminus of TRPV1 channel and L-ANAP methyl ester was incorporated into TRPV1 N terminus with a TAG amber stop codon mutation. The time course of fluorophore amplitude change was monitored accompanied with whole-cell patch clamping.

The imaging system was built on a Nikon TE2000-U microscope. The excitation light was generated by an Ar laser. The duration of light exposure was controlled by a computer-driven mechanical shutter (Uniblitz). A ×40 oil-immersion objective (NA 1.30) was used in these experiments. Spectral measurements were performed with an Acton SpectraPro 2150i spectrograph in conjunction with an Evolve 512 EMCCD camera. Both the shutter and the camera were controlled by MetaMorph software (Universal Imaging), synchronized with PatchMaster. In this way, the current and the spectral image were recorded simultaneously. From the spectral image, the eGFP/ANAP or eYFP/ANAP intensity ratio was calculated and used to follow the dynamic gating conformational changes during gating.

In the spectra FRET mode, two filter cubes (Chroma) were used. Cube I contained AT375/28× (excitation), ZT349rdc (dichroic), and ET500/40 m (emission), and cube II contained ZET488/10×, ZT458rdc, and ET510/80m. Two spectroscopic images were obtained from each cell: one with the ANAP excitation at 375 nm using cube I and the other with the eGFP/eYFP excitation at 490 nm using cube II. From these two images, the total emission spectrum was constructed.

**Molecular modeling**. To model the heat-induced desensitized state of TRPV1, membrane-symmetry-loop modeling was performed using the Rosetta molecular modeling[41] version 2015.25. As the heat-activated state of TRPV1 has not been experimentally resolved yet, we employed the DkTx- and RTX-bound open state (PDB ID: 3J5Q) as the starting structure, because previous studies have shown that these toxins activated the channel through heat activation pathway[26,27]. The selectivity filter, the pre S6 linker, and the S1–S2 linker were modeled de novo with the kinematic (KIC) loop modeling protocol[42,43]. Ten thousand to 20,000 models were generated each round. These models were first filtered by the SASA value at

the 630 site. Only the models with a decrease in SASA >20 Å$^2$ compared with the open state were allowed to pass. Among the filtered models, the top 20 models by energy were selected as the inputs for next round of loop modeling. After 14 rounds of KIC loop modeling, the top ten models converged well. The model with the lowest energy was finally selected as the desensitized state model. This model was further refined by the relax application[44] within the Rosetta suite.

To model the sidechain conformation of ANAP within the TRPV1 models, we generated the rotamer library of ANAP as described[25,45]. Briefly, the chemical structure of ANAP was optimized by Gaussian version 09[46]. Then the backbone-dependent rotamer library was generated by the MakeRotLib application[45] in Rosetta. Then the ANAP residue was incorporated into TRPV1 models in the open and desensitized states by the Backrub application[47,48] in Rosetta. For each state, 5000 models were generated and the model with the lowest energy was selected.

Command lines used in Rosetta to perform the modeling processes were attached in Methods. SASA of each residue in TRPV1 structures models was measured by RosettaScripts[49] within the Rosetta suite. The scripts to perform SASA measurements and filtering were also attached in Supplementary Methods.

Similarly, the interaction between distal N and C termini of TRPV1 was also modeled de novo with the KIC loop modeling protocol[42,43].

Pore radius of a TRPV1 model was calculated by the HOLE program[50] version 2.0[51].

All the molecular graphics of TRPV1 models were rendered by UCSF Chimera[52] software version 1.12[53].

**DRG neurons**. DRG neurons were acutely dissociated and maintained in a short-term primary culture according to procedures[54]. Briefly, after mice were killed by $CO_2$ inhalation and decapitated, vertebral column was opened by a dorsal approach. DRG neurons were collected using fine forceps from both sides of the spinal column and transferred into DMEM medium (GIBCO), and the spinal nerves were cut off using fine ophthalmological scissors and digested by 22% collagenase and 12% trypsin (m/m, Sigma) for 20 min. Following the wash of collagenase and trypsin, DRG neurons were incubated at 37 °C in 5% $CO_2$. Cells were used for electrophysiological experiments within the next 24 h.

**Histological analysis**. Following fixation by 10% formalin and dehydration by an increasing concentration of alcohol, paw materials were embedded in paraffin and sectioned to a thickness of 5 μm using a histocut (Leica). Sections of paw tissues were deparaffinized and rehydrated for hematoxylin and eosin staining.

**Immunoblotting**. Mice feet were cut into small pieces and homogenized using a T10 basic ULTRA-Turrax (IKA) in RIPA lysis buffer (R0278, Sigma) supplemented with protease inhibitors (P8340, Sigma) and phosphatase inhibitors (P5726, Sigma). Following a 20 min incubation step at 4 °C and clarified by centrifuging at 4 °C (15 min, 12,000 × $g$), the supernatants were collected. The protein concentrations of the supernatants were measured using the BCA assay kit (23225, Thermo). A total of 40 μg of the protein samples were boiled with SDS-polyacrylamide gel electrophoresis (PAGE) sample buffer (BL502A, Biosharp) for 5 min and then separated by 12% SDS-PAGE followed by electro-transfer onto polyvinylidene fluoride (PVDF) membranes (ISEQ00010, Millipore). After blocking with 5% bovine serum albumin (4240GR100, Biofroxx) dissolved in Tris-buffered saline containing 0.1% Tween-20 (TBST, 2.42 g L$^{-1}$ Tris-base, 8 g L$^{-1}$ NaCl, 0.1% Tween-20 pH 7.6) at room temperature for 1 h, the PVDF membranes were incubated overnight with primary antibodies at 4 °C. They were then washed for three times using TBST before incubating with secondary antibodies for 1 h at room temperature. After incubation, the membranes were washed again for three times with TBST, then exposed to chemiluminescent reagents, and images captured using an ImageQuant LAS 4000 instrument (GE HealthCare). The immunoblotting was performed using the following antibodies: Caspase-3 (8G10) Rabbit mAb (#9665, Cell Signaling) and Mouse β-Actin Antibody (sc-69879, Santa Cruz), which were used as the primary antibodies, and horseradish peroxidase-labeled anti-rabbit (#7074, Cell Signaling) or anti-mouse (#7076, Cell Signaling) antibodies were used as secondary antibodies.

**Animal behaviors**. For the tail-immersion assay, mice were immobilized in aluminum foil, which allowed free tail movement. The tip of the tail (one-third of the length) was immersed in a water bath maintained at 45 °C and the latency to withdrawal of the tail was determined. The tail was removed from the bath immediately after a nociceptive response to prevent acute tissue damage.

For the hot-plate assay, mice were individually confined in a Plexiglas chamber on a metal surface set at 45, 48, 50, and 52 °C, and the latency to a nociceptive response (licking or shaking of hind paws, jumping) was determined. Mice were removed from the hot plate after a nociceptive response to prevent tissue damage.

For the walking distance assay, mice were individually tracked for 60 min confined in a Plexiglas chamber on a metal surface set at 45 °C (CIB, Inc.), which consisted of a controlled and stable temperature plate on an aluminum floor. Tracking was performed using a video camera-based system (ZS Dichuang) and the walking distance was determined.

For the iterative hot-plate assay, mice were individually confined in a Plexiglas chamber on a metal surface set at 45 °C for 30 min with a 4 h interval and repeated

for ten times. After the 45 °C heat treatment, the paws of mice were used for histological analysis and inflammatory factors detection.

**RNA-sequencing.** After mice were killed by $CO_2$ inhalation, DRG neurons were isolated and kept in liquid nitrogen until further processing. RNA from DRG of both WT and p-*trpv1* male adult mice was isolated using RNeasy Mini Kit (Qiagen) according to the manufacturer's protocol. The sequencing libraries were constructed from the total RNA and were subjected to DSN normalization according to the protocol that is provided with the NEBNext Ultra RNA Library Prep Kit (New England Biolabs, Inc.), and each library was sequenced on a hiseq X ten according to the manufacturer's recommendations, generating 150 bp paired-end reads (NextOmics, Inc.). Reads were aligned using the open-source, splice-aware tool Tophat2 (v2.0.5). Read counts per feature (genes or exons) were done by StringTie (v1.3.0). Count data were analyzed to identify differentially expressed genes using DESeq2 (v1.01.1) at FDR 5%.

**Data analysis.** All experiments were independently repeated for at least three times. All statistical data are expressed as means ± SEM. Two-sided Student's *t*-test was applied to examine the statistical significance. Statistical significance was accepted at a level of $P < 0.05$. NS indicates no significant difference.

**Reporting summary.** Further information on research design is available in the Nature Research Reporting Summary linked to this article.

## Data availability

Data supporting the findings of this work are available within the paper and its Supplementary Information files. The source data underlying Fig. 1–6, Supplementary Figs. 1–6, and Supplementary Tables 1–9 are provided as three Source Data files. Sequencing data that support the findings of this study have been deposited in the NCBI Sequence Read Archive (SRA) and are accessible under the BioProject ID PRJNA531036, with the BIG Submission Genome Sequence Archive (GSA, accession number CRA001521), and are accessible under the BioProject ID PRJCA001355.

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

## Acknowledgements

This work was supported by funding from National Science Foundation of China (331372208), Chinese Academy of Sciences (XDB31000000 and QYZDJ-SSW-SMC012), and Yunnan Province (2015HA023) to R.L., and from the National Science Foundation of China (31640071 and 31770835), Chinese Academy of Sciences (Youth Innovation Promotion Association and "Light of West China" Program), and Yunnan Province (2017FB037 and 2018FA003) to S.Y. This work was also supported by funding from National Science Foundation of China (31741067 and 31800990) and the Young Thousand Talents Plan of China to F.Y., and from NIH (R01NS103954 and R56NS097906) to J.Z.

## Author contributions

L.L., Y.W. and B.L. conducted the majority of experiments including mutagenesis, patch-clamp recordings, and biochemical, physiological, and animal assays. L.X., P.M.K. and F.Y. contributed in the construction of TRPV1 models. R.L., S.Y., F.Y. and J.Z. prepared the manuscript. P.M.K., S.Y., L.L. and Y.W. participated in data analysis and manuscript writing. S.Y., F.Y. and R.L. conceived and supervised the project.

## Additional information

**Competing interests:** The authors declare no competing interests.

