## [Peer Review File · Nature Communications]

Reviewers' comments:

Reviewer #1 (Remarks to the Author):

This is an interesting study by Luo et al. that combines phylogenetic comparison of TRPV1 orthologs, biophysical analysis of channel function and structure, and mouse genetics to transfer novel heat desensitization properties into mouse TRPV1 from platypus TRPV1. The scope of this study and thoughtful discussion elevate its significance beyond that of "just another structure-function study". Overall, the study appears to be well done, although I cannot speak to the details of the biophysical analyses. The following specific criticisms did arise:

- 1) The manuscript should be extensively proofread for English grammar
- 2) Page 3, paragraph 1: The phylogeny analysis is not well described. What do the authors mean by "positively selected"?
- 3) Figure 1 – The temperature dependence of pV1 should be in a regular figure instead of a supplementary figure
- 4) The amino acids included in the overexpressed N- and C- termini of TRPV1 should be more precisely defined.
- 5) For figure 1F, a control peptide of similar length to Pep_30 should be used to address specificity.
- 6) Figure 1H – 15C is too low of a starting point for the SPR experiments, especially when it is the only temperature compared with 35C. A better comparison would be over a range from 30C to 40C, to determine whether the proposed interaction matches the temperature dependence of Ah or Dh.
- 7) Figure 2B – The heat stimuli used have a variable plateau phase, which makes interpretation of the desensitization characteristics of the mutants difficult. The experiments should be repeated with more consistent heat stimuli.
- 8) The authors should discuss their findings in comparison with the results of Vlachova et al. J Neurosci. 2003 Feb 15;23(4):1340-50 which implicated the C terminus of rat TRPV1 in sensitization in response to heat.

Reviewer #2 (Remarks to the Author):

TRPV1 ion channel are essential components of the heat sensing and homeostasis mechanism in mammals and other animals. Several gating process in these channels have remained poorly understood. The mechanism by which temperature gates these channels remains unknown and several observations seem to paint a controversial picture of this process. Although temperature activation has been more thoroughly characterized, heat-dependent desensitization has remained very poorly characterized, let alone understood.

This manuscript by Luo et al. presents data pertaining a possible mechanism for heat-dependent

inactivation of TRPV1 currents and takes advantage of a comparative approach to address this problem. Although some of the data is compelling, I find that the FRET measurements are at best preliminary.

The rest of the paper contains interesting data, but some of the results need to be carefully presented and discussed in light of the known literature.

Here are my recommendations regarding this manuscript.

As the authors mention, heat dependent desensitization is largely unexplored, however, a recent report (Sanchez-Moreno et al., eLife, June 2018) has presented a characterization of the phenomenon in rat TRPV1 and shows that activation and desensitization are coupled.

The authors mention that heat desensitization is common in mammalian channels and that the process does not affect capsaicin activation. However, Sanchez-Moreno et al., eLife, clearly showed that heat-desensitized channels are also unresponsive to capsaicin. Since the authors use the mouse TRPV1, do they think this is a characteristic that differs between TRPV1 orthologs?

In figure 2b the authors show that the $\Delta 10$ deletion does not undergo heat desensitization, however this data as presented is misleading and hard to interpret, since the temperature reached is barely 40 deg C and heat desensitization would be visible at higher temperatures. Please show data at temperatures comparable to the single-point mutants in the same figure.

Figure 2 C, D and E show that single-point mutants in the C-terminus slow down heat desensitization, with time constants near 40 s. It is asserted that the WT mTRPV1 has a slower 17 s time constant. However, this comparisons need to be made at the same temperature, since the rate of heat desensitization is steeply temperature-dependent. It is evident from the traces in figure 1A and figure 2 that the temperatures reached in each experiment are not the same.

My main concern is with the FRET data between ANAP and YFP. It is well established that the fluorescence of GFP and its variants (YFP) is very sensitive to temperature. Moreover, prodan and its derived dyes, ANAP included, are extremely temperature sensitive (Lakowics, Princ.

Fluorescence Spectroscopy). The authors have not included any controls for the effects of temperature on fluorescence intensity and/or spectrum of their dyes and no discussion is provided. In the interest of reproducibility, it should be mentioned what provisions were taken to account for changes in the value of the refraction index of the bath and the expected movements of the microscope parts as the bath is heated. This is a key part of the paper that needs to be dealt with, since major conclusions are derived from these data.

Reviewer #1 (Remarks to the Author):

This is an interesting study by Luo et al. that combines phylogenetic comparison of TRPV1 orthologs, biophysical analysis of channel function and structure, and mouse genetics to transfer novel heat desensitization properties into mouse TRPV1 from platypus TRPV1. The scope of this study and thoughtful discussion elevate its significance beyond that of “just another structure-function study”. Overall, the study appears to be well done, although I cannot speak to the details of the biophysical analyses. The following specific criticisms did arise:

Response: We deeply appreciate the reviewer’s comment to the interest and importance of this study. As discussed in detail below, we have performed additional experiments and thoroughly revised the manuscript to address the concerns raised by the reviewer.

Comments:

1) The manuscript should be extensively proofread for English grammar

Response: Thank you. We have carefully corrected grammar errors throughout the revised manuscript.

2) Page 3, paragraph 1: The phylogeny analysis is not well described. What do the authors mean by “positively selected”?

Response: We have added detailed description of our phylogeny analysis in the methods section of the revised manuscript. We have rephrased the statement as “As illustrated in Supplementary Fig. 1b, trpv1 gene of platypus’ genome is highlighted by using the branch-site model (p-value: 0.0123), suggesting that this gene may undergo molecular evolution to participate in environmental adaptation.”

3) Figure 1 – The temperature dependence of pV1 should be in a regular figure instead of a supplementary figure.

Response: The comparison of temperature dependence between mV1 and pV1 has been moved to Figure 1c.

4) The amino acids included in the overexpressed N- and C- termini of TRPV1 should be more precisely defined.

Response: We have now indicated the amino acid number in Figure 1f.

5) For figure 1F, a control peptide of similar length to Pep_30 should be used to address specificity.

Response: As suggested by the reviewer, we have added the control peptide Pep_30 during sustained heating and observed no desensitization of pV1 as opposed to the effect of peptide mC (Figure 1g). The results fully support the specificity of mC peptide to induce heat desensitization in pV1.

6) Figure 1H – 15C is too low of a starting point for the SPR experiments, especially when it is the only temperature compared with 35C. A better comparison would be over a range from 30C to 40C, to determine whether the proposed interaction matches

the temperature dependence of Ah or Dh.

Response: We appreciate the reviewer's comments. Due to hardware limitation in the SPR experiments, it was difficult to obtain robust SPR signals when the testing temperature was higher than 35 °C. Therefore, as suggested by the reviewer we have now performed additional SPR experiments under two more temperature points (30 and 33 °C) (Figure 1h). As demonstrated in the temperature-effect relationship experiments (Figure 1h and 1i), there is a significant increase in SPR signal when the temperature is elevated from 30 °C to 35 °C.

7) Figure 2B – The heat stimuli used have a variable plateau phase, which makes interpretation of the desensitization characteristics of the mutants difficult. The experiments should be repeated with more consistent heat stimuli.

Response: Thank you (and also reviewer #2) for these comments. We have optimized our temperature control devices so that in the additional experiments we conducted (Figure 2b-2e), the temperature variation in the plateau phase has been minimized to be less than 1 °C (Supplementary Figure 1f).

8) The authors should discuss their findings in comparison with the results of Vlachova et al. J Neurosci. 2003 Feb 15;23(4):1340-50 which implicated the C terminus of rat TRPV1 in sensitization in response to heat.

Response: We appreciate the reviewer's suggestion. We have compared and discussed this and our studies in the Discussion of the main text. Briefly, in Vlachova's study they suggest that the C terminus is important for heat activation of TRPV1, here we observed that the C terminus interacts with N terminus during heat desensitization, which is an additional critical role of the C terminus in regulating the activities of TRPV1 channel (Vlachova et al, JNS, 2003).

Reviewer #2 (Remarks to the Author):

TRPV1 ion channel are essential components of the heat sensing and homeostasis mechanism in mammals and other animals. Several gating process in these channels have remained poorly understood. The mechanism by which temperature gates these channels remains unknown and several observations seem to paint a controversial picture of this process. Although temperature activation has been more thoroughly characterized, heat-dependent desensitization has remained very poorly characterized, let alone understood.

This manuscript by Luo et al. presents data pertaining a possible mechanism for heat-dependent inactivation of TRPV1 currents and takes advantage of a comparative approach to address this problem. Although some of the data is compelling, I find that the FRET measurements are at best preliminary.

The rest of the paper contains interesting data, but some of the results need to be carefully presented and discussed in light of the known literature.

Here are my recommendations regarding this manuscript.

Response: We appreciate the reviewer's comments. In the revised manuscript, we have carefully addressed the concerns of this reviewer, as outlined below.

Comments:

As the authors mention, heat dependent desensitization is largely unexplored, however, a recent report (Sanchez-Moreno et al., eLife, June 2018) has presented a characterization of the phenomenon in rat TRPV1 and shows that activation and desensitization are coupled.

Response: Thank you for reminding us. We have cited this paper in the Discussion and rephrased our statement as “However, although the heat-induced *Dh* has been characterized (Sanchez-Moreno, et al, eLife, 2018), its underlying biophysical mechanisms and physiological significance remain largely unexplored, partially due to the challenge that *Ah* and *Dh* of most mammalian TRPV1 channels are tightly entangled.” in the Introduction. In addition to the phenomenon of heat desensitization as described in the eLife study, we have thoroughly investigated the underlying molecular and structural mechanisms, as well as the physiological significance of this desensitization process. We believe that our findings have sufficiently advanced our knowledge of heat desensitization in TRPV1.

The authors mention that heat desensitization is common in mammalian channels and that the process does not affect capsaicin activation. However, Sanchez-Moreno et al., eLife, clearly showed that heat-desensitized channels are also unresponsive to capsaicin. Since the authors use the mouse TRPV1, do they think this is a characteristic that differs between TRPV1 orthologs?

Response: We rephrased the description as “The *Ah* current of pV1 is robust compared to the current activated by saturating capsaicin (Fig. 1a and Supplementary Table 1). Please note that we used saturated capsaicin (50 μ M) to activate the heat-desensitized mV1 (Supplementary Fig. 1c), because desensitized rTRPV1 channels also became insensitive to 4 μ M capsaicin (Sanchez-Moreno et al)”.

Indeed, we realized that the activation of heat-desensitized channels is required for much higher concentrations of capsaicin than that of TRPV1 channels in the resting state.

In figure 2b the authors show that the C Δ 10 deletion does not undergo heat desensitization, however this data as presented is misleading and hard to interpret, since the temperature reached is barely 40 deg C and heat desensitization would be visible at higher temperatures. Please show data at temperatures comparable to the single-point mutants in the same figure.

Response: We have performed new patch-clamp experiments with the C Δ 10 mutant at a stable temperature around 46 $^{\circ}$ C (Figure 2b), which is comparable to the temperature levels achieved in other experiments (Figure 2c to 2e, Supplementary Figure 1f). Similar to our previous results, we observed no heat desensitization in the C Δ 10 mutant even at this elevated temperature. Our observation is also consistent with the previous report that heat desensitization is eliminated when the distal C terminus is deleted (Joseph J et al., JBC, 2013).

Figure 2 C, D and E show that single-point mutants in the C-terminus slow down heat desensitization, with time constants near 40 s. It is asserted that the WT mTRPV1 has a slower 17 s time constant. However, this comparisons need to be made at the same temperature, since the rate of heat desensitization is steeply temperature-dependent. It

is evident from the traces in figure 1A and figure2 that the temperatures reached in each experiment are not the same.

Response: Thanks for pointing it out. As in the response to Reviewer 1's comment #7, we have first optimized our temperature control device to make the temperature in the plateau phase more stable. In the additional experiments we conducted (Figure 2b-2e), the temperature variation in the plateau phase has been minimized to be less than 1 °C. With better temperature control, we have now conducted new heat desensitization experiments at the similar plateau temperature 45 ± 2 °C (Supplementary Figure 1f), which makes the comparison of heat desensitization time courses between different mutants more reliable (Figure 2b-2e).

My main concern is with the FRET data between ANAP and YFP. It is well established that the fluorescence of GFP and its variants (YFP) is very sensitive to temperature. Moreover, prodan and its derived dyes, ANAP included, are extremely temperature sensitive (Lakowics, Princ. Fluorescence Spectroscopy). The authors have not included any controls for the effects of temperature on fluorescence intensity and/or spectrum of their dyes and no discussion is provided. In the interest of reproducibility, it should be mentioned what provisions were taken to account for changes in the value of the refraction index of the bath and the expected movements of the microscope parts as the bath is heated. This is a key part of the paper that needs to be dealt with, since major conclusions are derived from these data.

Response: We appreciate the reviewer's comments. We have conducted new tests to verify the FRET results. To measure and correct the effect of heating on fluorophores and therefore FRET efficiency, we first directly incorporated ANAP to the N terminus of the GFP/YFP to replace the second amino acid. We use these constructs as negative controls. When heated to 45 °C, we observed that the YFP/ANAP and GFP/ANAP fluorescence intensity ratio was decreased (Supplementary Figure 3a-d). Such a decrease in intensity ratio suggests that the ratio increase observed in our experiment between YFP and ANAP (Figure 1b) should be even larger than the values our previously measured. Based on such changes in negative controls, we have corrected our YFP/ANAP or GFP/ANAP intensity ratio measurements (Figure 1). The corrected ratios clearly demonstrated that there was a larger than originally measured increase in FRET efficiency in heat desensitized mV1, while there was no such a change in pV1 where heat desensitization process was absent. In addition, we also observed, that as correctly pointed out by the reviewer, though the fluorescence intensity of ANAP decreased upon heating, the emission spectrum of ANAP did not shift in our tested temperature range (Supplementary Figure 3e and 3f).

To further consolidate our FRET experiments, we took an alternative experimental strategy where we measured the changes in FRET induced by centipede toxin RhTx. Previous studies (Yang S et al., 2015, Nature Communications; Bae C et al., eLife, 2016) have shown that peptide toxins like RhTx and DkTx mimic heat activation of TRPV1. Indeed, we observed that RhTx not only activates mV1 but also desensitize mV1 upon perfusion, but in pV1 where the heat desensitization is absent RhTx also failed to desensitize the channel though activation could be observed (Figure 3e). Therefore we believe that the prolonged application of RhTx mimics the effect of sustained heat. Using RhTx as a tool, we observed that just like the heat, RhTx induces significant changes in the YFP/ANAP ratio (therefore FRET efficiency) in mV1, but not in pV1 lacking the *Dh*

process (Figure 3f and 3g). In summary, based on our observation on the ANAP-YFP/GFP constructs and RhTx induced FRET changes, we believe that the FRET increase observed upon sustained heating truly reflects the conformational rearrangements during the *Dh* process in mV1.

REVIEWERS' COMMENTS:

Reviewer #1 (Remarks to the Author):

The authors have nicely addressed all of this reviewer's comments, and the manuscript is substantially improved. No further criticisms.

Reviewer #2 (Remarks to the Author):

The authors have addressed my main concerns and I am glad to see the new experimental evidence supports the paper's claims, specially regarding the FRET data. This is a nice advance in understanding gating in TRPV1 ion channels.

REVIEWERS' COMMENTS:

Reviewer #1 (Remarks to the Author):

The authors have nicely addressed all of this reviewer's comments, and the manuscript is substantially improved. No further criticisms.

Response: The authors sincerely thank this reviewer.

Reviewer #2 (Remarks to the Author):

The authors have addressed my main concerns and I am glad to see the new experimental evidence supports the paper's claims, specially regarding the FRET data. This is a nice advance in understanding gating in TRPV1 ion channels.

Response: The authors appreciate this reviewer for this highly encouraging comment.